# A Convergent Mixed-Methods Evaluation of a Co-Designed Evidence-Based Practice Module Underpinned by Universal Design for Learning Pedagogy

**DOI:** 10.3390/nursrep15070236

**Published:** 2025-06-27

**Authors:** Stephanie Craig, Hannah McConnell, Patrick Stark, Nuala Devlin, Claire McKeaveney, Gary Mitchell

**Affiliations:** School of Nursing and Midwifery, Queen’s University Belfast, Medical Biology Centre (MBC), 97 Lisburn Road, Belfast BT9 7BL, Northern Ireland, UK; h.mcconnell@qub.ac.uk (H.M.); p.stark@qub.ac.uk (P.S.); nuala.devlin@qub.ac.uk (N.D.); c.mckeaveney@qub.ac.uk (C.M.)

**Keywords:** evidence Based Practice, pedagogy, nursing education, education, universal design for learning

## Abstract

**Background**: The concept of evidence-based practice (EBP) is globally relevant in current healthcare climates. However, nursing students and teachers often struggle with integrating EBP effectively into a curriculum. This has implications for the way students learn to use evidence for their nursing practice. A new undergraduate EBP module was co-designed with current nursing students and university staff throughout 2023. Underpinning the module was a UDL (universal design for learning) pedagogy consisting of flexible approaches to learning for nursing students which included co-developed videos, co-developed audio podcasts, and co-developed serious games to complement traditional flipped classroom learning. The module commenced in September 2023, running in Year 1 one of a 3-year undergraduate nursing program, and was co-taught by staff and senior students. **Methods**: A pre/post-test design was used to collect data on student attitude, knowledge, and utilization of EBP. A total of 430 students completed two validated questionnaires, the EBP Beliefs Scale© and EBP Implementation Scale©, before and after the module. Following the post-test, six focus group interviews were also conducted with 58 students to explore how the module informed student nursing practice whilst attending clinical placement during Year 1. A convergent mixed-methods design was employed. Sample attrition occurred (~25%). Effect sizes and 95% confidence intervals were calculated for primary outcomes. **Results:** Quantitative data was analyzed using paired t-tests and this highlighted statistically significant improvements in attitude, knowledge and utilization of evidence-based practice after learning (*p* < 0.001). Qualitative data was transcribed verbatim, thematically analyzed, and highlighted three main findings; EBP is my business, EBP positively influenced the care of my patients and EBP has positively impacted my professional development. **Conclusions**: Partnership with current nursing students in the co-design and implementation of a module about EBP was associated with improvements in student knowledge, attitude and utilization of evidence in practice. These factors are likely to also improve professional competence and ultimately patient care.

## 1. Introduction

Evidence-based practice (EBP) is ‘…an approach to healthcare that utilizes the most current research available in order to improve the health and safety of patients while reducing overall costs and variation in health outcomes…’ [1]. EBP is a systematic process designed to make improvements in healthcare [2], becoming the gold standard of healthcare practice [3] capable of improving patients’ safety, the quality of care and reducing costs by addressing ineffectiveness in healthcare settings [4,5]. This approach aims to ensure that healthcare professionals use the best scientific evidence when making informed decisions as well as designing and implementing healthcare interventions [6,7,8]. The process of EBP was endorsed by the United Kingdom’s nursing regulator, the Nursing and Midwifery Council (NMC) [9] as a professional responsibility of registered nurses and an embedded component of undergraduate nursing degrees [10]. Nursing education has sought to prepare students for EBP competency through the creation of frequent and relevant opportunities to ask clinically relevant questions and search scientific and clinical databases through the critical appraisal of research [11,12]. However, there is no single EBP competency framework guiding educators in the application of EBP in nursing education. The existence of several EBP competency frameworks has contributed to variation in EBP teaching, disparities in EBP competencies and reduced confidence in EBP implementation [5,12,13].

Studies exploring the experiences of EBP nursing education suggest nursing students struggle to link theory and evidence [14]. Despite acknowledging EBP contributes to better outcomes for patients, nursing students also report being intimidated and overwhelmed by EBP [15]. Nursing professionals express a ‘double standard’; a high expectation to carry out EBP whilst experiencing significant barriers to accessing and understanding EBP literature [16,17]. Whilst Hines et al. [17] suggest the biggest challenge to EBP implementation are human factors such as fear and anxiety which have a significant influence on EBP learning outcomes, examination performance as well as future EBP implementation [18,19,20,21]. Conversely, research has demonstrated developing more positive attitudes toward EBP significantly increases the likelihood of future application of EBP [22].

Pedagogical strategies acknowledge the importance of designing interventions to improve students’ understanding and interest in EBP [23]. Unlike VAK/VARK (Visual, auditory, read/write and kinesthetic) learning styles, the Universal Design for Learning (UDL) suggests a novel approach that aims to provide greater educational opportunities for all learners and is thought to have increased potential in nursing education. This purposeful and integrative framework considers the needs of a diverse range of learners [24,25,26] and is distinguished from other approaches due to its grounding in neuroscientific aspects of learning [27]. The Centre for Applied Special Technology (CAST) crucially define UDL as a framework “to improve and optimize teaching and learning for all people based on scientific insights into how humans learn” [28].

While historically, UDL has deep connections with the provision of disability education, due to its inclusive approach it is now recognized to have suitability as a comprehensive pedagogy which can transform teaching and learning for all students [29]. UDL aims to optimize teaching practice by providing insight into how students learn [28] with the overarching aim of creating barrier free, universal access to education for all students [25,30,31,32]. UDL offers diverse modalities in which to support learner development, by acknowledging that learning is not a static but rather a continuous process, which needs to be tailored to the individual [25]. The three core principles of UDL (a) providing learners with multiple means of engagement, by stimulating their interests and motivating them to learn, (b) providing multiple means of representation, presenting content through multiple modalities, such as audio, visual, and kinesthetic, and (c) providing multiple means of expression, creating a variety of activities by which students can demonstrate their learning [28,33] are embedded within the module. Through the three core UDL principles, students can consider the why, the what, and the how of learning [28,34].

Establishing UDL as a process-based framework rather than an intervention allows greater flexibility and adaptability for each learner, as it incorporates variation to approaches to learning and does not assume a default position of one size fits all [29]. Active learning draws on many activities including multiple intelligences that are central to this is the ability to bridge the theory to practice gap [35]. UDL offers space for the learner to be an active participant and enables nursing students to develop the skills necessary to apply evidence-based practice [23]. Therefore, the aim of this study was to co-design an evidence-based practice (EBP) module informed by the Universal Design for Learning (UDL) framework, and to evaluate its impact on first-year undergraduate nursing students’ beliefs about and implementation of EBP. The objectives of the study were threefold: (1) to assess changes in students’ EBP beliefs and implementation before and after participating in the module using validated measurement tools; (2) to explore students’ experiences of the module and how it influenced their application of EBP during clinical placements; and (3) to examine the potential of UDL as a pedagogical approach to improve engagement, confidence, and competence in EBP. By integrating UDL principles into the design and delivery of an EBP module, this study seeks to address existing disparities in EBP education and contribute toward a more inclusive, effective, and evidence-informed approach to undergraduate nursing education.

## 2. Materials and Methods

### 2.1. Design

The pre-test–post-test study adopted a pre–post evaluation with embedded qualitative inquiry using non-probability convenience sampling of year-one undergraduate student nurses undertaking a compulsory evidence-based nursing module at a large teaching university in the United Kingdom. The sample was derived from September 2022 and February 2023 teaching cohorts. Students anonymously completed two validated scales (Evidence-Based Practice Beliefs Scale© (EBPB), and Evidence-Based Practice Implementation Scale© (EBPI)) in a study by Melnyk et al. [36] online via Microsoft Forms prior to commencing the module and upon completion of the module. The EBPB scale contains 16 items which are scored on a five-point Likert scale, ranging from strongly disagree (1) to strongly agree (5). The EBPI scale consists of 18 items, each rated on a 5-point frequency scale reflecting how often students implemented each item over the past 8 weeks: 0 = 0 times, 1 = 1–3 times, 2 = 4–5 times, 3 = 6–8 times, and 4 = more than 8 times. Permission to use these scales for this evaluation was obtained in June 2023 by the authors who originally developed the instruments. On completion of the module, all year-one undergraduate students were invited to participate in a 30 min focus group interview with a member of the research team to discuss how the module helped to inform their nursing practice during clinical placements.

This study employed a convergent parallel mixed-methods design, following the Good Reporting of A Mixed-Methods Study (GRAMMS) framework (https://www.equator-network.org/reporting-guidelines/the-quality-of-mixed-methods-studies-in-health-services-research/ (accessed on 2 May 2024). A mixed-methods approach was selected to provide an evaluation of the co-designed module, capturing both measurable changes in student outcomes and explanatory data about their experiences. Quantitative and qualitative data were collected concurrently: students completed validated pre- and post-intervention questionnaires, and a subset participated in focus group interviews following the module. Quantitative data were collected using the EBP Beliefs Scale© and EBP Implementation Scale©, while qualitative data were gathered through six online focus group interviews. All Year 1 students enrolled in the module (n = 601) were eligible for the quantitative component and focus group participants (n = 58) were self-selected via voluntary sign-up, reflecting a non-probability sampling approach for both components. The priority of data types was equal, with neither strand considered dominant. Integration of data occurred during the interpretation phase through side-by-side comparison of quantitative results and qualitative themes, highlighting convergences and divergences. Quantitative data were analyzed using descriptive statistics, paired t-tests, and correlation analyses in SPSS (v27), while qualitative data were thematically analyzed following Braun and Clarke’s six-phase approach. This design enabled a more robust understanding of the impact of the module than could be achieved by a single method alone.

### 2.2. Module Co-Design

The development of the “Evidence-Based Nursing 1” module followed a systematic seven-step co-design approach, underpinned by the principles of accelerated co-design methodology [37]. Accelerated co-design, grounded in participatory and human-centered design principles, supports the rapid development of effective and contextually relevant educational interventions by engaging end-users and stakeholders throughout the design and refinement process [38,39,40]. This approach was selected to ensure the module was both pedagogically robust and grounded in the lived experiences of nursing students, nurse educators, and healthcare practitioners, while also facilitating timely development aligned with institutional academic calendars. The co-design team comprised 34 stakeholders, including 25 year-two undergraduate nursing students who had completed a previous version of the module, five nurse educators responsible for delivering the module, two patient and public involvement (PPI) representatives, and two practicing registered nurses. Students were selected based on successful completion of the previous EBP module and their voluntary expression of interest in contributing to module redesign. PPI contributors were identified through the university’s stakeholder engagement panel. Workshop activities included structured discussions, persona development, low-fidelity prototyping, and prioritization matrices. The diversity of stakeholders provided rich, authentic data across learner, educator, patient, and clinical practice perspectives. The design process took place over an eight-week period and included structured workshops, asynchronous feedback loops, and collaborative development activities. Early design workshops identified shared learning outcomes, with a high-level goal of supporting nursing students to develop a working knowledge of EBP and its relevance to safe, high-quality nursing care. Stakeholders emphasized that EBP must be differentiated from nursing research; while not all nurses would conduct research, all were expected to engage with evidence in practice. Therefore, the module prioritized introducing students to the concept of EBP as an essential nursing competency and aimed to build confidence in using and applying evidence within placement settings. The final co-designed module focused on six core topic areas agreed upon by stakeholders:What is Evidence-Based Practice (EBP)?—Introducing the principles, relevance, and practical importance of EBP in nursing.Understanding Literature Reviews—Teaching students how to interpret and use, rather than conduct, literature reviews to inform clinical decisions.Qualitative Research—Introducing qualitative methodologies and supporting students to interpret findings in relation to patient experience and care quality.Quantitative Research—Supporting students to interpret statistical and outcome-based evidence to inform practice decisions.Ethics and Research Governance—Exploring ethical considerations in research and service evaluation, and their implications for safe, patient-centered care.Quality Improvement and Practice Development—Introducing quality improvement (QI), clinical audit, and service development as methods for translating evidence into practice.

In line with the Universal Design for Learning (UDL) framework, stakeholder feedback revealed a wide range of learner preferences and needs, particularly among first-year students who often found EBP intimidating. A key design recommendation was to incorporate motivational strategies that would support engagement without overwhelming learners. Students consistently reported that while the concept of EBP felt abstract and challenging at first, they became increasingly comfortable once immersed in the learning. To support this progression, the module included mandatory, incentivized learning elements alongside flexible, self-directed activities. Each of the six core topics was delivered via a one-hour face-to-face lecture, which was mandatory and delivered as part of the formal timetable. These lectures formed the foundation of the module. Complementing this core delivery, three additional learning activities were available for each topic, tailored to diverse learning styles:Bespoke Reading Resources: Three open-access, pre-selected readings were curated for each topic. These were selected for accessibility and relevance, with students able to access them directly via hyperlinks. This eliminated the need for students to locate literature independently in Year 1, thereby reducing barriers to engagement while building familiarity with academic sources.Student Co-Designed Audio Podcasts: A 30 min podcast accompanied each topic, co-designed and co-delivered by former students who had successfully completed the module. These “on-the-go” learning tools were not intended for notetaking, but to support reflection and consolidation during everyday activities like commuting or exercising. Each episode concluded with a live “quiz-off” among student contributors, aligned with the final module assessment (a multiple-choice question exam), enhancing relevance and motivation.Kinesthetic Learning Activities: These varied by topic and included reflective writing tasks, multiple-choice quizzes, and interactive module-hosted discussions (e.g., X chat forums). These were entirely optional and designed for flexible engagement.

Importantly, students recommended that participation in learning should be recognized through assessment. As a result, 20% of the final module grade was allocated to engagement with these optional activities. The learning management system automatically tracked student activity, and these data contributed to their final grade. This incentive structure was proposed by students themselves and aimed to foster a sense of reward and motivation for engaging with the learning material. In the final weeks of the module, students requested a Consolidation of Learning Pack to help them review content before their final assessment. They did not want this labeled as a “revision” resource but instead framed as an opportunity to revisit and reflect on their learning. This pack included:A two-hour optional masterclass was delivered live by the module lead, which offered a 15-minute summary and a 5-minute Q&A for each topic.A mock MCQ exam aligned with final assessment content.A final podcast summarizing key learning points and their relevance to nursing practice, developed by student champions.A co-designed serious digital game featuring over 300 MCQ-style questions to support knowledge application and exam preparation in a flexible, mobile-friendly format.

Through this comprehensive, co-designed development process, the Evidence-Based Nursing 1 module was transformed into a dynamic, multimodal educational intervention tailored to the needs, expectations, and realities of first-year nursing students, grounded in evidence, and informed by stakeholder expertise.

### 2.3. Participants

The participants were all year-one undergraduate nursing students (adult, mental health, learning disability and children’s fields) in September 2022 and February 2023 cohorts (n = 601) undertaking the Evidence-Based Nursing 1 module at the recruiting university. Students were informed that their participation in the study was voluntary and would not affect their module grade and those who chose to participate were not incentivized to do so.

### 2.4. Data Collection

Data collection occurred at two points for each respective cohort, the first prior to commencing the Evidence-Based Nursing 1 module (September 2022 and February 2023) and the second on completion of the module at the end of Year 1 (August 2023 and December 2023). Students were informed of the project via email by their year lead who was not associated with the project. The pre-questionnaire was distributed to students one week prior to the module commencing, alongside an information sheet and contact details of the research team. The post-questionnaire was made available to students one week before the module ended. Students were able to access the pre-test questionnaire for two weeks, the link was removed thereafter as students were actively engaging in the module. The post-test questionnaire was available to students during the final week of the module and two weeks after the module ended, the link was available to all students, including those who did not complete the pre-test questionnaire. Participant responses were collected via Microsoft Forms. Before accessing the pre-test and/or post-test questionnaire, students were required to confirm, through an online consent form, that they had read the participant information sheet and agreed to participate. This confirmation served as their digital signature of consent. Written (paper-based) consent was not feasible due to the large cohort size and the online nature of the data collection process. Students were invited to self-register for a focus group interview slot during the final week of module teaching. Eventbrite was used to enable students to select a date/time convenient to them. Upon registration, students received an information sheet, and written consent was obtained on the day of the focus group interview.

### 2.5. Tools Used

Demographic data including specialism studied (Adult nursing, Children and Young People nursing, Learning Disability nursing or Mental Health nursing), age and gender was collected from all participants. Students were asked to complete two scales: Evidence-Based Practice Beliefs Scale© (EBPB) and Evidence-Based Practice Implementation Scale© (EBPI) [36]. The validity and reliability of these scales has been demonstrated with Spearman–Brown and Cronbach alpha reliability coefficients >0.85 for both scales [36].

### 2.6. Ethical Considerations

This study was approved by the investigating school’s research ethics committee (MHLS20_94) in July 2022. Students received written information about the study one week before the module commenced and contact details of the research team. The same process was followed for the post-questionnaire, students received information one week prior to the module ending. Students were informed: participation was voluntary; involvement in the study would not affect their marks in the evidence-based nursing module; personal details would be anonymized and they could withdraw at any time. Students were emailed a link to pre-test and post-test questionnaires via Microsoft Forms in which participants needed to confirm they had read the participant information sheet and consent to participate prior to completing the questionnaire. Written paper-based consent was obtained from focus group participants prior to interview commencement. Audio recordings were stored securely and anonymized by removing identifying details during transcription.

### 2.7. Data Analysis

Quantitative data was collected and automatically collated via Microsoft Forms, quantitative descriptive statistics were utilized to identify and interpret key findings from the study. Student numbers were used to directly compare pre-test and post-test results and were anonymized after the data was paired, then transferred to SPSS for statistical analysis, the data was deleted from Microsoft Forms following this. Qualitative data was digitally audio-recorded, and recordings were saved on a password-protected server at the investigating university. The audio recordings were transcribed verbatim and thematically analyzed [41]. Personal information that had the potential to identify participants was not saved in the recording.

## 3. Results

This study completed data collection and analysis of the cohort of students undertaking the Evidence-Based Nursing 1 module in September 2022 and February 2023. Data collection and analysis of results took place in January 2024 using the Evidence-Based Practice Beliefs Scale© (EBPB) and Evidence-Based Practice Implementation Scale© (EBPI) [36]. Pre-test data was collected in September 2022 and February 2023 (total number of responders = 579), and post-test data was collected in June 2023 and December 2023 (total number of responders = 430). This study followed a convergent parallel mixed-methods design. Quantitative and qualitative data were collected concurrently and analyzed independently, then merged during interpretation to generate data about the module’s impact. Three research questions (RQ) guided the study.
RQ1. Does attitude towards EBP change significantly during an EBP module for first-year student nurses?RQ2. Do levels of EBP change significantly during an EBP module for first-year student nurses?RQ3. Does attitude towards EBP and level of EBP measured at the end of the EBP module correlate with EBP examination scores?

### 3.1. Demographics

Demographic data was collected from all participants who completed both the pre- and post-questionnaires. This included information on gender, age, nursing specialism, and nationality. Table 1 provides a summary of the demographic characteristics of the final paired sample (n = 430), allowing for assessment of the sample’s representativeness.

### 3.2. Analysis Methodology

Scores for the 16-item attitude scale and the 18-item practice scale were calculated as total scores, summing each item to represent the positivity of their attitude and the frequency of their EBP practice, respectively. A higher score on the attitude scale represents a more positive attitude towards EBP, and a higher score on the practice scale represents a higher frequency of using EBP in practice. Students were asked to enter a unique identifier at both pre- and post-assessment points, enabling accurate matching of responses. Only those with valid matches across both timepoints were included in the paired analyses. The workbook scores are analyzed as the raw scores, gathered from end of year examinations. All analyses were carried out using SPSS version 27. Descriptive statistics were calculated for pre-test and post-test scores for attitude and practice scales and for the workbook score. Distribution of pre-test attitude and practice scores were examined using histograms to investigate any floor or ceiling effects or potential for regression to the mean when comparing with post-test scores. Two paired t-tests were conducted, to examine change from pre-test to post-test for attitude and practice, and two correlation analyses were carried out, to examine relationships between attitude/practice and workbook score.

Two comparisons and two correlational analyses were conducted during this study—pre-test to post-test changes for attitude and practice and correlations between attitude and workbook score and practice and workbook score. A Bonferroni correction was applied to the alpha value when determining the statistical significance of the results of these analyses to reduce the risk of false positives associated with multiple comparisons [42]. Although Bonferroni adjustment is often used in confirmatory analyses, we acknowledge that its application in exploratory educational research may be overly conservative. However, given the limited number of primary comparisons (n = 4), the correction was applied to minimize the risk of Type I error. Alpha (0.05) was divided by the total number of analyses in this study (4) to give an alpha value of a = 0.0125. Results of the pairwise comparisons in this study were, therefore, only considered to be statistically significant if their associated *p*-value was 0.0125 or below.
RQ1. Does attitude towards EBP change significantly during an EBP module for first-year student nurses?

The distribution of attitude scale pre-test scores (Figure 1) shows that there is an approximately normal distribution, with no strong positive or negative skew or evidence of ceiling or floor effects.

Descriptive statistics (Table 2) show that post-test scores (M = 64.73, SD = 5.58) were higher than pre-test scores (M = 58.36, SD = 6.88). The difference between these two scores was statistically significant as indicated by the paired samples t-test which gave a result of t(429) = 16.29, *p* < 0.001. In answer to RQ1, yes, attitudes towards EBP significantly improved during the EBP module.
RQ2. Do levels of EBP change significantly during the course of an EBP module for first-year student nurses?

The distribution of practice scale pre-test scores (Figure 2) shows a floor effect, i.e., a high frequency of participants scoring at the minimum score (18) for EBP at the beginning of the first-year EBP module.

Descriptive statistics (Table 3) show that post-test scores (M = 31.74, SD = 8.24) were higher than pre-test scores (M = 20.35, SD = 3.50). The difference between these two scores was statistically significant as indicated by the paired samples t-test which gave a result of t(429) = 27.50, *p* < 0.001. In answer to RQ2, yes, levels of EBP significantly increased during the course of the EBP module.
RQ3. Does attitude towards EBP and level of EBP measured at the end of the EBP module correlate with EBP examination scores?

A significant moderately positive correlation was found between workbook score and attitude towards EBP (r(410) = 0.29, *p* < 0.001), i.e., those with a more positive attitude scored more highly on the workbook examination. No significant correlation was found between workbook score and practice (r(411) = 0.006, *p* = 0.907).

### 3.3. Qualitative

In total, six focus group interviews were conducted with 58 first-year undergraduate nursing students who had completed the co-designed Evidence-Based Nursing 1 module. These interviews were held via Microsoft Teams between one and four weeks after the conclusion of the module to ensure participants had the opportunity to apply their learning in clinical settings. Each participant had undertaken three clinical placements during the academic year, and the interviews were scheduled to occur after the final placement to maximize reflection on real-world application. Focus groups lasted between 35 and 50 min and were facilitated online for accessibility and convenience. All discussions were audio-recorded, transcribed verbatim, and thematically analyzed following Braun and Clarke’s six-phase framework [41]. Two researchers [SC, GM] independently coded the transcripts and met regularly to reconcile coding frameworks, ensuring coder triangulation. Saturation was deemed to be reached after the sixth focus group. NVivo software (v14) was used to facilitate coding and theme development. Thematic analysis resulted in the development of three overarching themes that captured students’ experiences and perceptions of evidence-based practice (EBP) after completing the module. The first theme, ‘EBP is my business,’ explored how students shifted from feeling uncertain about EBP to seeing it as a fundamental and ongoing part of their professional identity, both as students and future registered nurses. The second theme, ‘EBP positively influenced the care of my patients,’ examined how students actively applied EBP principles and resources during clinical placements, resulting in noticeable improvements to patient care, while also navigating the challenges of implementing evidence within hierarchical practice environments. The final theme, ‘EBP positively impacted my academic and professional development,’ highlighted how the module changed students’ approaches to learning, boosted their confidence, and fostered a deeper sense of ownership over their education and emerging professional roles.

### 3.4. Theme 1: EBP Is My Business

Many students entered the module with unclear or limited understandings of evidence-based practice (EBP), often perceiving it as abstract, research-heavy, or not directly relevant to the clinical role of a student nurse. For many, the concept was conflated with academic research or scientific study, which contributed to a sense of alienation from the topic. As one participant described, “At the start I thought this was like a research module, like stats and all that…I was honestly a bit lost with what it meant for us” {FG2, P3}. Students consistently noted how the module was initially more difficult to connect with because, unlike anatomy or clinical skills, the impact of EBP on patient care was not immediately tangible. This sense of unfamiliarity contributed to some initial disengagement: “I didn’t get why it was important; it didn’t seem to relate to real nursing or being on the ward” {FG1, P9}.

However, this initial uncertainty was gradually replaced by growing confidence and clarity as the module progressed. Students began to see how EBP intersected with everyday clinical decisions, helped them critically interpret the information they encountered, and empowered them to contribute meaningfully to discussions around patient care. A key turning point came with the clarification that EBP was distinct from academic research, it was not about producing original studies, but about being able to find, understand, and apply existing evidence. “I didn’t realise it wasn’t about doing research. It’s more about being able to look at research and go, is this good or not?” {FG4, P7}. This demystification of EBP helped to reposition the concept as a core nursing skill, not just an academic exercise. Students described how the module “broke it down” and made EBP approachable: “Once I understood the triad, you know, the patient, clinician, the evidence…then I started seeing it all around me on placement” {FG6, P2}.

As their confidence grew, students began to identify EBP as an essential part of their current and future professional roles. The module’s use of relatable examples, tools like PRISMA diagrams, and exposure to resources like JBI gave them a framework for assessing the quality of evidence. Many participants said they no longer felt overwhelmed when encountering journal articles or clinical guidelines. One student noted, “I still don’t understand all the stats, but now I can read an RCT and get the gist…like sample sizes, is it real-world, that sort of thing” {FG3, P6}. Importantly, they recognized that EBP was not just their business in isolation, it was something to be shared and championed with others in practice. Several students reflected on their growing sense of advocacy and responsibility for creating evidence-informed cultures: “It’s my business because I need to use the evidence, but it’s also about helping others to do the same. Like even as a student I’ve shared guidelines with HCAs and nurses” {FG5, P4}. This theme ultimately illustrates a shift in mindset, with students coming to view EBP as a core part of their identity, not as optional, but integral to being a safe, effective, and accountable nurse.

### 3.5. Theme 2: EBP Positively Influenced the Care of My Patients

The practical relevance of EBP was most clearly realized when students applied learning from the module during clinical placements. Over the course of the academic year, participants experienced three placements across different settings and these included hospital wards, community health services and care homes, all of which offered opportunities to apply EBP in varied contexts. Students described how the module helped them to use guidelines, interpret research summaries, and link theory to practice. One student shared, “I was with a dementia patient who was getting upset, and because of EBP I tried reminiscence therapy instead of asking for sedation. It really worked” {FG2, P6}. This use of evidence to guide compassionate, person-centered care was echoed across many focus groups.

Students also identified and addressed poor or superficial uses of evidence in practice. One particularly impactful example came from a care home placement, where a student observed the use of music therapy being reduced to simply playing the radio. Drawing on module learning, the student explained that music therapy should be interactive, reflective, and led by a trained facilitator. “I talked to the team and showed them what proper music therapy is. We changed how it was done, and it helped. I even got an award for it” {FG4, P2}. Such accounts demonstrate the capacity of undergraduate students to influence practice positively when empowered with the right tools. Other students described sourcing research to inform communication strategies for people with learning disabilities, or ways to prepare children for the operating theater, again showing EBP’s immediate impact.

However, students also encountered barriers when trying to influence clinical practice. In some cases, their suggestions based on evidence were dismissed by more senior staff. One student recounted trying to raise concerns about polypharmacy in a palliative patient: “I mentioned guidelines say to reduce non-essential meds, but the GP just said no, it’s fine. It was really frustrating, and I didn’t really feel seen” {FG6, P4}. Another reflected on a care assistant giving excessive salt to a patient with heart failure, despite known risks: “They just said, ‘he wants it’, and that was that. It was against the evidence, but I didn’t feel I could push it because I am still a student” {FG3, P5}. These stories suggest that while the module helped students identify good evidence and think critically about practice, more focus is needed on how to effectively challenge and influence clinical decisions, particularly when faced with resistance or hierarchy.

Despite these challenges, the prevailing sentiment was that EBP strengthened students’ confidence in clinical settings. Many reported using evidence to validate their care decisions or support the introduction of new practices. One powerful example came from a student who noted restrictive practices being used inappropriately in a dementia unit. After identifying evidence showing such interventions increased fall risk, the student shared a quality improvement idea with the team: “They were using a table to stop him falling, but it was making him angry and didn’t help. I shared a paper with the charge nurse, and they changed it, and it worked better” {FG1, P7}. Such instances highlight how student nurses, when equipped with evidence and confidence, can become drivers of safer, more ethical care. EBP, in this sense, was not just a theory because students used it, shared it, and saw it change lives.

### 3.6. Theme 3: EBP Positively Impacted My Academic and Professional Development

As students began to understand EBP’s relevance to patient care, they also described how the module transformed their academic engagement and professional identity. Many students initially approached university-level learning with a passive mindset, expecting lectures to be sufficient. However, the module’s design, with accessible reading materials, podcasts, and gamified quizzes, encouraged deeper, self-directed learning. “I used to just go to lectures and revise from notes. But for this, I listened to the podcast, did the quizzes, read the links. I saw the difference in my grade and started doing the same in other modules” {FG5, P5}. This theme reflects a broader shift in learning behavior because students did not just consume knowledge; they started curating it.

Students expressed appreciation for the variety of learning tools made available, with many highlighting the podcasts as especially influential. “I had never listened to a podcast for learning before. I was dreading it. But once I heard it, I was like, this is actually really good. It wasn’t boring at all” {FG1, P8}. The informal tone, peer-led delivery, and quiz-off style were all cited as motivating features. Others commented on the simplicity and usefulness of direct-access readings: “Other modules give you a reading list and say ‘go find it’. But this one just gave you the links. It saved time and made it easier to actually read the articles” {FG2, P2}. The module’s digital game with over 300 MCQs was also praised: “I loved the game. It was fun and actually helped with the exam. I wish all modules had something like that” {FG6, P6}.

Crucially, the module did not just build knowledge; it helped students reflect on their learning preferences and take more control of their academic development. Several participants described how they had started to ask other module leads for resources like podcasts or revision masterclasses after experiencing their benefits in this module. Others said it gave them permission to explore content on their own terms: “Before this, I just did what we were told. Now I look up extra stuff if I’m interested. It made me realise I can be curious and that helps” {FG4, P9}. This self-awareness and confidence suggest that the module achieved more than its stated outcomes. It contributed to a culture of lifelong learning, supported students in understanding how they learn best, and empowered them to transfer these approaches to other parts of their education.

## 4. Discussion

The co-design process that underpinned the development of the EBP module demonstrates the potential of participatory curriculum design in higher education. By involving 34 stakeholders, including current and former nursing students, nurse educators, practicing clinicians, and patient representatives, the module was shaped to directly reflect the needs, challenges, and priorities of its intended users. This approach aligns with the growing body of literature advocating co-design in educational interventions, which has repeatedly demonstrated that such collaborative efforts yield curricula that are more relevant, engaging, and effective [42,43,44]. The accelerated co-design methodology used here facilitated rapid yet meaningful engagement, allowing for iterative feedback and authentic stakeholder ownership [38,39,40]. Notably, the decision to allocate 20% of the module’s assessment to engagement with optional, student-designed activities emerged directly from student input, reinforcing the motivational power of co-design [45]. Quantitative results from this study, which included statistically significant improvements in students’ attitudes, knowledge, and implementation of EBP (*p* < 0.001), mirror findings from other co-designed interventions, which have consistently resulted in enhanced learner confidence and practical application of knowledge [46,47]. Qualitative data further highlighted how students felt empowered to question established practices and integrate evidence into patient care, confirming that participatory design not only improves educational outcomes but also fosters professional growth and critical thinking [48,49]. Integration of quantitative and qualitative data occurred through side-by-side comparison in the discussion. Future reporting may benefit from joint displays to better visualize convergence or divergence across data strands.

Pedagogically, the module was underpinned by Universal Design for Learning (UDL), a framework that is increasingly recognized for its capacity to create inclusive, flexible, and engaging learning environments in higher education [50,51]. UDL’s three core principles, (1) Representation, (2) Action and Expression, and (3) Engagement, were embedded throughout the module, offering students a variety of ways to access content and demonstrate understanding [52]. For example, each topic was delivered through a mandatory face-to-face lecture, complemented by co-designed podcasts, curated reading resources, and kinesthetic activities, all tailored to accommodate diverse learning preferences and needs. This approach is particularly significant for neurodiverse learners, who often face barriers in traditional educational settings [53,54]. By prioritizing flexibility and individualized learning pathways, the module contributed to building capacity in higher education, equipping students with the skills and self-awareness necessary for lifelong learning and professional adaptability. The integration of asynchronous, blended learning was another cornerstone of the module’s success. Recognizing the demands of nursing students, who often juggle academic, clinical, and personal responsibilities, the module offered a blend of in-person lectures and on-demand digital resources [55,56].

The acceptability and impact of the module were strongly affirmed by both quantitative and qualitative findings. Statistically significant improvements were observed in students’ EBP beliefs and implementation scores. These gains exceeded those reported in comparable studies and were complemented by qualitative data. Focus group participants described tangible changes in their clinical practice, including increased confidence in questioning outdated protocols and a greater propensity to seek out and apply research evidence. Notably, many participants reported accessing academic literature during placements, an important shift given the well-documented barriers to EBP engagement in nursing education [57,58,59]. These findings suggest that the module’s approach shows promising evidence in bridging the theory-practice gap and fostering a culture of inquiry and evidence-informed care.

Integration of quantitative and qualitative data occurred during the interpretation phase of the study. Quantitative results indicating statistically significant improvements in EBP attitudes and implementation scores were examined alongside qualitative findings that explored students lived experiences of using evidence in practice. This side-by-side comparison enabled a deeper understanding of how observed changes manifested in real-world clinical contexts. For example, increased implementation scores aligned with focus group accounts describing confident use of guidelines and evidence-based decision-making. Where quantitative findings showed no correlation between practice scores and exam outcomes, qualitative data provided context, suggesting external barriers to applying EBP during placements. These points of integration demonstrate how the two datasets enriched and explained one another, providing a more comprehensive evaluation of the module’s impact.

Taken together, these findings demonstrate that a well-structured, co-designed EBP module underpinned by inclusive pedagogical principles can serve as a powerful mechanism for cultural and professional transformation in undergraduate nursing education. Beyond improving knowledge and skill, the module cultivated a shift in students’ sense of professional identity, helping them to see themselves as active contributors to evidence-informed care. The alignment of co-design and UDL principles not only enhanced learner engagement and outcomes but also modeled best practices in curriculum development, emphasizing collaboration, accessibility, and responsiveness to real-world demands. As healthcare systems increasingly expect newly qualified nurses to navigate complex clinical environments with confidence and criticality, embedding EBP education early in the curriculum, through participatory, flexible, and contextually grounded approaches offers a promising strategy for preparing a more resilient, reflective, and evidence-literate nursing workforce. Future research should explore the longitudinal impact of such modules on graduate practice and investigate how these pedagogical models can be scaled and adapted across diverse nursing and allied health programs.

## 5. Strengths and Limitations

A key strength of this study is its mixed-methods design, which combined validated quantitative tools with qualitative focus groups. The co-design approach ensured that the module was contextually grounded and learner-informed, enhancing its relevance and acceptability. The use of the UDL framework supported diverse learning needs and facilitated high engagement across a large, multidisciplinary student cohort. Additionally, the study evaluated real-world impact by including data collected after students completed clinical placements. However, several limitations must be acknowledged. The absence of a control group limits causal attribution, as all students were required to complete the module. Additionally, the absence of a control or comparison group limits causal inference. It is possible that the observed changes could be attributed to extraneous variables such as maturation or external academic influences. Future studies should incorporate a randomized or matched control group to better isolate the intervention effect. Response rates for post-test data were lower than pre-test, potentially introducing non-response bias. Approximately 25% of students did not complete the post-test. A comparison of demographic characteristics between pre- and post-responders was not conducted, which limits the ability to assess non-response bias. Future evaluations should perform sensitivity analyses to account for attrition. Focus groups were self-selected, and participants may have had more positive perceptions, furthermore, focus group participants were self-selected, which may have attracted more engaged or motivated students. Some groups contained up to 10 participants, potentially limiting the depth of individual contributions during discussions, which could skew findings. Furthermore, outcomes were measured only in the short term; longer-term follow-up would be needed to assess sustained changes in EBP behaviors and attitudes. The moderate correlation between attitude and workbook score (r = 0.29) should not be interpreted as evidence of the module’s effectiveness, as confounders were not controlled for. The absence of correlation with practice frequency warrants further investigation and could reflect contextual barriers to applying EBP in clinical placements. Finally, the study employed non-probability convenience sampling, which introduces potential for selection bias and limits generalizability. As all Year 1 students were recruited without randomization, findings may not be fully representative of broader undergraduate nursing populations.

## 6. Conclusions

This study provides strong evidence that a co-designed, pedagogically EBP module was associated with enhancements in undergraduate nursing students’ confidence, knowledge, and application of EBP. Statistically significant improvements were observed in students’ attitudes and practice of EBP, and qualitative data illustrated how students transferred their learning into clinical settings. The integration of the UDL framework contributed to an inclusive, accessible, and motivating learning environment, while the participatory development process empowered students and educators alike. These findings suggest the potential value of embedding co-designed EBP modules early in nursing curricula to foster evidence-informed care and critical thinking. As healthcare systems increasingly demand competent, inquiry-driven practitioners, such approaches may hold promise for preparing nurses who are not only knowledgeable but also confident in implementing EBP in practice. Future research should examine the long-term effects of such modules on clinical performance, practice culture, and professional development in early-career nursing.

## Figures and Tables

**Figure 1 nursrep-15-00236-f001:**
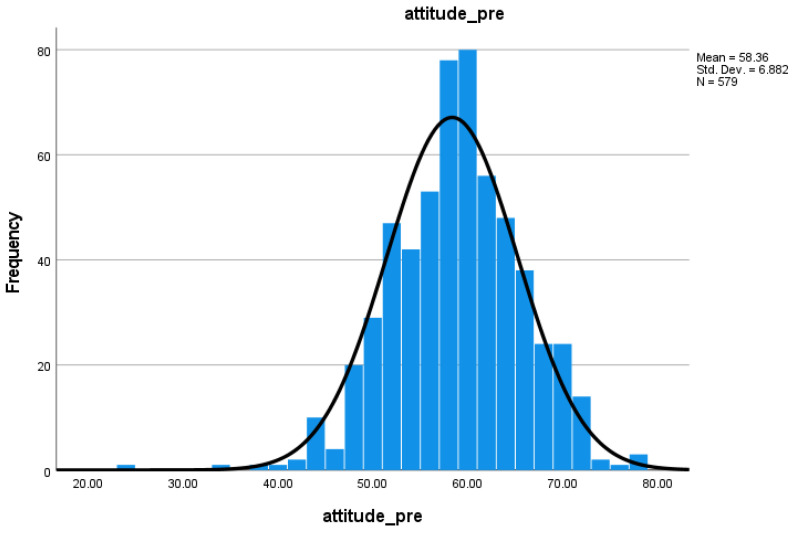
Distribution of pre-test scores for attitude scale.

**Figure 2 nursrep-15-00236-f002:**
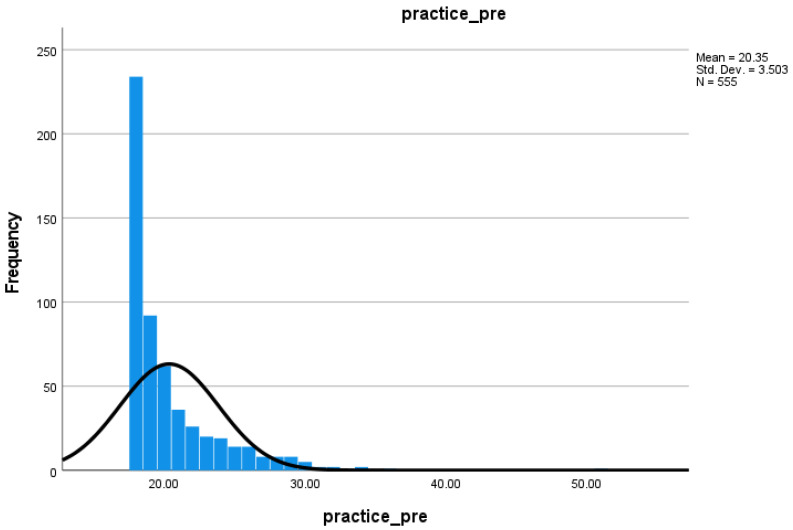
Distribution of pre-test scores for practice scale.

**Table 1 nursrep-15-00236-t001:** Participant Demographics.

Characteristic	Category	n	%
Gender	Female	391	91%
	Male	39	9%
Age Group	18–25 years	310	72%
	26–35 years	77	18%
	36–45 years	25	6%
	46–54 years	18	4%
	55+ years	0	0%
Field of Nursing	Adult	258	60%
	Children’s	86	20%
	Mental Health	52	12%
	Learning Disability	34	8%
Nationality/Ethnicity	UK and Ireland	413	96%
	International	17	4%

**Table 2 nursrep-15-00236-t002:** Descriptive statistics for pre-test and post-test attitude scores.

	N	Minimum	Maximum	Mean	Std. Deviation
Pre-test attitude	579	24.00	78.00	58.36	6.88
Post-test attitude	430	45.00	78.00	64.73	5.58

**Table 3 nursrep-15-00236-t003:** Descriptive Statistics for pre-test and post-test practice scores.

	N	Minimum	Maximum	Mean	Std. Deviation
Pre-test practice	555	18.00	51.00	20.35	3.50
Post-test practice	430	18.00	68.00	31.74	8.24

## Data Availability

Data is contained within the article. The original contributions presented in this study are included in the article. Further inquiries can be directed to the corresponding authors.

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
