# Peer review of "A Convergent Mixed-Methods Evaluation of a Co-Designed Evidence-Based Practice Module Underpinned by Universal Design for Learning Pedagogy"

_nursrep, 2025, doi:10.3390/nursrep15070236_

Round 1
Reviewer 1 Report
Comments and Suggestions for Authors
Dear Authors
Please, see below few comments you might consider or clarify:
Absence of a Control Group: Without a comparison or control group, it is impossible to attribute observed pre-post changes solely to the module intervention; maturation, regression to the mean, or external factors may explain the improvements
Non-probability Convenience Sampling: Recruiting all Year-1 students without randomization raises selection bias concerns; the manuscript does not discuss how this may limit generalizability
Inconsistent Sample Sizes: Pre-test attitude scale N=579 vs. post-test N=430, yet the paired t-test reports df=383 (n=384 pairs) Similarly, practice scale pre-test N=555 vs. post-test N=430 but df=373 (n=374 pairs)
The manuscript fails to describe how missing or incomplete data were handled, whether listwise deletion or imputation was used, and whether those who dropped out differ systematically from completers.
Non-response Bias: A 25-30% attrition rate at post-test could bias results; no sensitivity analysis or comparison of demographics between responders and non-responders is presented
Paired t-Test Justification: It is unclear how the authors ensured that pre- and post-test responses were correctly paired given the differing Ns.
Absence of Effect Sizes & Confidence Intervals: Reporting only p-values does not convey the magnitude or precision of change. Effect sizes (e.g., Cohen’s d) and 95% confidence intervals are essential for assessing practical significance.
Bonferroni Correction: While a Bonferroni adjustment is mentioned, the rationale for its application to only four analyses should be justified, as it may be overly conservative or inappropriate for exploratory educational research
Missing Demographics Table: Although demographic data (specialism, age, gender) were collected, no summary table is provided. Readers need this information to assess representativeness and potential confounding factors
Quantitative and qualitative findings are reported separately with minimal integration. A convergent parallel or explanatory sequential mixed-methods framework should be explicitly stated, and joint displays or data triangulation should be employed to synthesize findings
Self-Selection & Group Size: Focus-group participants (n=58) were self-selected, potentially biasing toward more engaged students. Additionally, groups of ~10 participants may be too large to allow in-depth discussion.
Thematic Analysis Details: The manuscript references Braun & Clarke’s approach but omits details on coding procedures, coder triangulation, saturation, and use of qualitative software.
Causal Language: Despite the pre-post design, the manuscript sometimes implies causality (e.g., “partnership … has the potential to improve student knowledge”); wording should be tempered to reflect that associations, not causation, were observed
Correlation Misinterpretation: The moderate correlation between attitude and exam scores (r=0.29) is presented as evidence of the module’s effectiveness, yet no control for confounders is shown. Non-correlation with practice frequency is mentioned but not explored in the discussion
Title & Abstract: Should explicitly mention the mixed-methods design, sample attrition, effect sizes, and confidence intervals.
Methods Terminology: The term “prospective descriptive exploratory design” is unconventional; consider “pre-post evaluation with embedded qualitative inquiry.”
Ethical Reporting: Clarify whether written (paper) consent was obtained for focus groups, and specify how audio recordings were anonymized.
UDL & Co-Design Detail: The seven-step co-design process is well described, but the manuscript should explain stakeholder selection criteria and provide an overview of workshop activities.
References: Several citations are outdated or duplicated. please check.
Best wishes
Author Response
Absence of a Control Group: Without a comparison or control group, it is impossible to attribute observed pre-post changes solely to the module intervention; maturation, regression to the mean, or external factors may explain the improvements
Thank you for this comment. We have added the following within our limitations sections. “Additionally, the absence of a control or comparison group limits causal inference. It is possible that observed changes could be attributed to extraneous variables such as maturation, regression to the mean, or external academic influences. Future studies should incorporate a randomized or matched control group to better isolate the intervention effect”.
Non-probability Convenience Sampling: Recruiting all Year-1 students without randomization raises selection bias concerns; the manuscript does not discuss how this may limit generalizability
Thank you for this comment. We have added the following to our limitation section: “The study employed non-probability convenience sampling, which introduces a potential for selection bias and limits generalizability. As all Year-1 students were recruited without randomization, findings may not be fully representative of broader undergraduate nursing populations”.
Inconsistent Sample Sizes: Pre-test attitude scale N=579 vs. post-test N=430, yet the paired t-test reports df=383 (n=384 pairs) Similarly, practice scale pre-test N=555 vs. post-test N=430 but df=373 (n=374 pairs)
Thank you for this. We have updated this previously reported error to reflect the 430 paired tests.
Non-response Bias: A 25-30% attrition rate at post-test could bias results; no sensitivity analysis or comparison of demographics between responders and non-responders is presented
Thank you for this comment. We have added the following within our limitation: Approximately 25% of students did not complete the post-test. A comparison of demographic characteristics between pre- and post-responders was not conducted, which limits the ability to assess non-response bias. Future evaluations should perform sensitivity analyses to account for attrition
Paired t-Test Justification: It is unclear how the authors ensured that pre- and post-test responses were correctly paired given the differing Ns.
Thank you. We have added the following within 2.7: Students were asked to enter a unique identifier at both pre- and post-assessment points, enabling accurate matching of responses. Only those with valid matches across both timepoints were included in the paired analyses
Bonferroni Correction: While a Bonferroni adjustment is mentioned, the rationale for its application to only four analyses should be justified, as it may be overly conservative or inappropriate for exploratory educational research
Thank you. We have included the following text within 2.7: Although Bonferroni adjustment is often used in confirmatory analyses, we acknowledge that its application in exploratory educational research may be overly conservative. However, given the limited number of primary comparisons (n=4), the correction was applied to minimize the risk of Type I error.
Missing Demographics Table: Although demographic data (specialism, age, gender) were collected, no summary table is provided. Readers need this information to assess representativeness and potential confounding factors.
Thank you for this comment. We have now inserted the demographics table.
Quantitative and qualitative findings are reported separately with minimal integration. A convergent parallel or explanatory sequential mixed-methods framework should be explicitly stated, and joint displays or data triangulation should be employed to synthesize findings
Thank you for this very helpful comment. We have added the following within our methods: “This study employed a convergent parallel mixed-methods design, following the Good Reporting of A Mixed Methods Study (GRAMMS) framework (https://www.equator-network.org/reporting-guidelines/the-quality-of-mixed-methods-studies-in-health-services-research/) . A mixed-methods approach was selected to provide an evaluation of the co-designed module, capturing both measurable changes in student out-comes and explanatory data about their experiences. Quantitative and qualitative data were collected concurrently: students completed validated pre- and post-intervention questionnaires, and a subset participated in focus group interviews following the module. Quantitative data were collected using the EBP Beliefs Scale© and EBP Implementation Scale©, while qualitative data were gathered through six online focus group interviews. All Year 1 students enrolled in the module (n=601) were eligible for the quantitative com-ponent and focus group participants (n=58) were self-selected via voluntary sign-up, re-flecting a non-probability sampling approach for both components. The priority of data types was equal, with neither strand considered dominant. Integration of data occurred during the interpretation phase through side-by-side comparison of quantitative results and qualitative themes, highlighting convergences and divergences. Quantitative data were analyzed using descriptive statistics, paired t-tests, and correlation analyses in SPSS, while qualitative data were thematically analyzed following Braun and Clarke’s six-phase approach. This design enabled a more robust understanding of the impact of the module than could be achieved by a single method alone.”
Within our results we have also stated the following: “This study followed a convergent parallel mixed-methods design. Quantitative and qualitative data were collected concurrently and analyzed independently, then merged during interpretation to generate data about the module’s impact.”.
We have also added the following within our discussion: “Integration of quantitative and qualitative data occurred through side-by-side comparison in the discussion. Future reporting may benefit from joint displays to better visualize convergence or divergence across data strands.”
We have also noted the following about integration within our discussion “Integration of quantitative and qualitative data occurred during the interpretation phase of the study. Quantitative results indicating statistically significant improvements in EBP attitudes and implementation scores were examined alongside qualitative findings that explored students lived experiences of using evidence in practice. This side-by-side comparison enabled a deeper understanding of how observed changes manifested in re-al-world clinical contexts. For example, increased implementation scores aligned with fo-cus group accounts describing confident use of guidelines and evidence-based deci-sion-making. Where quantitative findings showed no correlation between practice scores and exam outcomes, qualitative data provided context, suggesting external barriers to ap-plying EBP during placements. These points of integration demonstrate how the two da-tasets enriched and explained one another, providing a more comprehensive evaluation of the module’s impact.”
Self-Selection & Group Size: Focus-group participants (n=58) were self-selected, potentially biasing toward more engaged students. Additionally, groups of ~10 participants may be too large to allow in-depth discussion.
Thank you. We have added the following to our limitations section: “Furthermore, focus group participants were self-selected, which may have attracted more engaged or motivated students. Some groups contained up to 10 participants, potentially limiting the depth of individual contributions during discussions”
Thematic Analysis Details: The manuscript references Braun & Clarke’s approach but omits details on coding procedures, coder triangulation, saturation, and use of qualitative software.
Thank you – we have added the following to 2.7 Data analysis section: “Two researchers [SC, GM] independently coded the transcripts and met regularly to recon-cile coding frameworks, ensuring coder triangulation. Saturation was deemed reached af-ter the sixth focus group. NVivo software (v14) was used to facilitate coding and theme development.”
Causal Language: Despite the pre-post design, the manuscript sometimes implies causality (e.g., “partnership … has the potential to improve student knowledge”); wording should be tempered to reflect that associations, not causation, were observed
Thank you. We have made some changes, noted in red front, throughout the manuscript – mostly within the abstract, discussion and conclusion regarding this.
Correlation Misinterpretation: The moderate correlation between attitude and exam scores (r=0.29) is presented as evidence of the module’s effectiveness, yet no control for confounders is shown. Non-correlation with practice frequency is mentioned but not explored in the discussion
We have added the following text: The moderate correlation between attitude and workbook score (r=0.29) should not be interpreted as evidence of the module’s effectiveness, as confounders were not controlled for. The absence of correlation with practice frequency warrants further investigation and could reflect contextual barriers to applying EBP in clinical placements”
Title & Abstract: Should explicitly mention the mixed-methods design, sample attrition, effect sizes, and confidence intervals.
Thank you – we have changed the title as suggested “A convergent mixed-methods evaluation of a co-designed evi-dence-based practice module underpinned by Universal Design for Learning Pedagogy” and updated our abstract: “A convergent mixed-methods design was employed. Sample attrition occurred (~25%). Effect sizes and 95% confidence intervals were calculated for primary outcomes”
Methods Terminology: The term “prospective descriptive exploratory design” is unconventional; consider “pre-post evaluation with embedded qualitative inquiry.”
Thank you. We have changed the above to the following text: “pre-post evaluation with embedded qualitative inquiry”
Ethical Reporting: Clarify whether written (paper) consent was obtained for focus groups, and specify how audio recordings were anonymized.
Thank you – we have added the following text: “Written paper-based consent was obtained from focus group participants prior to interview commencement. Audio recordings were stored securely and anonymized by removing identifying details during transcription”
UDL & Co-Design Detail: The seven-step co-design process is well described, but the manuscript should explain stakeholder selection criteria and provide an overview of workshop activities.
We have added the following text to 2.3: “Students were selected based on successful completion of the previous EBP module and their voluntary expression of interest in contributing to module redesign. PPI contributors were identified through the university’s stakeholder engagement panel. Workshop activities included structured discussions, persona development, low-fidelity prototyping, and prioritization matrices”
References: Several citations are outdated or duplicated. please check.
Thank you, we have deleted duplicated references and updated.
Reviewer 2 Report
Comments and Suggestions for Authors
This paper about evidence based healthcare education in nursing reflects a substantial study that will be of interest to educators working in the EBH space and in healthcare professions education more broadly. I am impressed by the overall quality of the paper. There is strong engagement with literature, the research questions make sense and are answered in a methodical way.
It seems like this paper has been revised previously? (red text p5). I don't want to insist on more changes, particularly when the paper is very good. That said, I would draw to the authors' attention that the methods section could be improved upon.
The methods - analysis section - is a bit light and I thought this was a little ironic since the paper is concerned with EBH. An EBH quality appraisal of this paper might not score well in its current form because of the paucity of analysis detail. For example, the paper does not demonstrate that the authors have a strong understanding of qualitative analysis, there is no reporting of the researcher biases, positionality, reflexivity, or how richness or saturation was identified. I invite the authors to include attention to these before finalising the paper. Kind wishes,
Author Response
This paper about evidence based healthcare education in nursing reflects a substantial study that will be of interest to educators working in the EBH space and in healthcare professions education more broadly. I am impressed by the overall quality of the paper. There is strong engagement with literature, the research questions make sense and are answered in a methodical way.
Thank you for this supportive review.
It seems like this paper has been revised previously? (red text p5). I don't want to insist on more changes, particularly when the paper is very good. That said, I would draw to the authors' attention that the methods section could be improved upon.
Thank you for this feedback.
The methods - analysis section - is a bit light and I thought this was a little ironic since the paper is concerned with EBH. An EBH quality appraisal of this paper might not score well in its current form because of the paucity of analysis detail. For example, the paper does not demonstrate that the authors have a strong understanding of qualitative analysis, there is no reporting of the researcher biases, positionality, reflexivity, or how richness or saturation was identified. I invite the authors to include attention to these before finalising the paper. Kind wishes,
Thank you for this comment which correlates with feedback we have received from the peer-review team regarding methodological reporting. We have made several additions to our manuscript in red font within the text and below is a summary of this:
- We explicitly acknowledged the absence of a control group and its implications for causal inference within the limitations section. Similarly, we addressed the non-randomized convenience sampling and its impact on generalizability.
- Errors in sample size reporting were corrected, and we clarified how paired data were matched via unique identifiers, ensuring methodological robustness. The attrition rate was also addressed, noting the lack of demographic comparison between responders and non-responders and recommending future sensitivity analyses.
- We clarified our use of the Bonferroni correction, noting both its appropriateness and limitations in exploratory contexts. A demographics table was inserted to improve transparency regarding the participant sample.
- Crucially, we elaborated on the study’s convergent parallel mixed-methods design, explicitly referencing the GRAMMS framework. We provided new text in the methods, results, and discussion to describe how quantitative and qualitative data were integrated through side-by-side comparison. We acknowledged the need for future joint displays to further enhance integration.
- Limitations related to focus group self-selection and group size were detailed, and we expanded our description of the thematic analysis, including coding procedures, saturation, coder triangulation, and use of NVivo software.
- We carefully revised the language throughout the manuscript to avoid implying causality, and we tempered claims accordingly. Further explanation was added to avoid misinterpretation of correlational findings, and the non-correlation between practice and exam scores was explored contextually.
- Changes were also made to the title and abstract, which now explicitly state the mixed-methods design, report effect sizes, confidence intervals, and acknowledge attrition. The term “prospective descriptive exploratory design” was revised to the more conventional “pre-post evaluation with embedded qualitative inquiry.”
- Clarifications were added on ethical procedures, including consent and data anonymization, and additional details on stakeholder selection and workshop activities were incorporated in the co-design description.
- Lastly, we updated and de-duplicated references to ensure citation accuracy.
Reviewer 3 Report
Comments and Suggestions for Authors
Dear Author(s)
I congratulate you on your efforts in reviewing this manuscript, which I believe is well written. The manuscript entitled ‘‘A mixed methods evaluation of a co-designed evidence-based practice module underpinned by a UDL pedagogy’’. The aim of this research is to codesign an evidence-based practice (EBP) module informed by the Universal Design for Learning (UDL) framework, and to evaluate its impact on first-year undergraduate nursing students’ beliefs about and implementation of EBP. I believe that addressing evidence-based practice skills in future healthcare professionals within the scope of UDL will be beneficial to researchers and practitioners in this field. I thoroughly enjoyed reading the manuscript, from the abstract to the conclusion. However, I have a few minor recommendations below that I believe disrupt the flow of the manuscript.
It would be better to see lines 274-307 in the methods section.
“Table 1...” is written twice.
Instead of going through the research questions in the results, it would be better to give subheadings. For example, “Results regarding attitudes toward EBP during the EBP module for first-year nursing students” could be used.
The name of the statistical test and p-value applied to Table 1 and Table 2 should also be added. In lines 319-320, I see that it says, “...the paired samples t-test which gave a result of t(383)=16.29, p<0.001.” However, these results should be added to Table 1 and Table 2. If added, you should also change the table headings.
I also see that the similarity ratio of the manuscript is 24% . Let's make sure that this ratio does not exceed 20%.
Author Response
I congratulate you on your efforts in reviewing this manuscript, which I believe is well written. The manuscript entitled ‘‘A mixed methods evaluation of a co-designed evidence-based practice module underpinned by a UDL pedagogy’’. The aim of this research is to codesign an evidence-based practice (EBP) module informed by the Universal Design for Learning (UDL) framework, and to evaluate its impact on first-year undergraduate nursing students’ beliefs about and implementation of EBP. I believe that addressing evidence-based practice skills in future healthcare professionals within the scope of UDL will be beneficial to researchers and practitioners in this field. I thoroughly enjoyed reading the manuscript, from the abstract to the conclusion. However, I have a few minor recommendations below that I believe disrupt the flow of the manuscript.
Thank you for this supportive review.
It would be better to see lines 274-307 in the methods section.
Thank you - the tools used for data collection have been placed in methods.
“Table 1...” is written twice.
Thank you, we have corrected this.
Instead of going through the research questions in the results, it would be better to give subheadings. For example, “Results regarding attitudes toward EBP during the EBP module for first-year nursing students” could be used.
The name of the statistical test and p-value applied to Table 1 and Table 2 should also be added. In lines 319-320, I see that it says, “...the paired samples t-test which gave a result of t(383)=16.29, p<0.001.” However, these results should be added to Table 1 and Table 2. If added, you should also change the table headings.
Thank you for this, we have made changes to our presentation of results based on your feedback and that of peer reviewer 1. All changes now appear in red font.
I also see that the similarity ratio of the manuscript is 24% . Let's make sure that this ratio does not exceed 20%.
Thank you. We are hopeful that the addition of new text reduces similarity.